# Quality of antenatal care and outcomes of Hypertensive Disorders in Pregnancy among antenatal attendees: A comparison of urban and periurban health facilities in Ghana

**Pauline Boachie-Ansah**[1¤]*, **Berko Panyin Anto**[1], **Afia Frimpomaa Asare Marfo**[1], **Edward Tieru Dassah**[2,3], **Ivan Eduku Mozu**[1], **Joseph Attakora**[1,4]

1 Faculty of Pharmacy and Pharmaceutical Sciences, Department of Pharmacy Practice, Kwame Nkrumah University of Science and Technology, Kumasi, Ghana, 2 Department of Population, Family and Reproductive Health, School of Public Health, Kwame Nkrumah University of Science and Technology, Kumasi, Ghana, 3 Department of Obstetrics and Gynaecology, Komfo Anokye Teaching Hospital, Kumasi, Ghana, 4 Department of Internal Medicine, Komfo Anokye Teaching Hospital, Kumasi, Ghana

¤ Current address: Department of Pharmacy Practice, Private Mail Bag, Kwame Nkrumah University of Science and Technology, Kumasi, Ghana

* paulineboachie.ansah@gmail.com

**Data Availability Statement:** All relevant data are within the paper and its Supporting Information

## Abstract

Rural-urban-peri urban disparity assessments on health outcomes have been considered as critical determinants of health and health service outcomes. It is policy relevant in terms of the burden of disease and also provides focus on target interventions. This study aimed to assess the differences in the quality of Ante-natal Care (ANC) and the outcomes of Hypertensive Disorders in Pregnancy (HDPs) from selected health facilities in Ghana. This was a questionnaire-based cross-sectional study. Data on demographics, proportions of HDPs, quality of ANC and the outcomes of HDPs were collected. Logistic regression models were used to examine the association of the independent variables with the location of the health of facility. A total of 500 pregnant women were included in this study. There were 270 (54%) urban and 230 (46%) peri-urban dwellers. The proportion of HDPs varied with the location of the health facility. Women attending urban health facilities were more likely to be hypertensive ($\mu^2 = 126.4$; p<0.001), have chronic hypertension with superimposed pre-eclampsia (p< 0.001), have good quality ANC ($\mu^2 = 41.28$; p< 0.001), deliver full term ($\mu^2 = 4.83$; p = 0.028), and have excellent knowledge on HDPs ($\mu^2 = 227.65$; p< 0.001) compared to women receiving care in peri-urban health facilities. The method of delivery and outcome of birth did not statistically vary amongst the periurban and urban health facilities. There was an increase in the proportion in preterm in urban compared to periurban. The burden of HDPs was high in urban health facilities with high proportion of its mothers receiving quality ANC as well as having excellent knowledge on HDPs compared to mothers receiving care at the periurban health facilities. There is a need to target maternal care interventions to the periurban health facilities to improve obstetric health outcomes.

files. The dataset analysed for this study is available at the figshare data repository. Link to data 352 repository https://figshare.com/articles/dataset/Plos_One_data-_PBoachieAnsah_dta/23978928.

**Funding:** PBA received a local scholarship from the Ghana National Petroleum Corporation (GNPC) Foundation, Ghana (GNPC/FDN/C4/KNUST/18) Sponsors played no role in the study design or manuscript www.gnpcghana.com/scholarship.html The funders had no role in study design, data collection and analysis, decision to publish, or preparation of the manuscript.

**Competing interests:** The authors have declared that no competing interests exist.

# Introduction

Health disparity refers to a significant difference in the health outcomes such as prevalence, morbidity, mortality, survival, hazard, of a defined population to another population which could be a reference population [1]. Several factors have been identified to account for health disparities [2–4]. These factors are broadly categorized as demographic characteristics [5], health-related behaviors [6], socioeconomic factors and environmental risks [7]. The underlying mechanisms of how these factors account for the variations are very complex and not fully understood.

Globally, rural-urban-peri urban disparity assessments on health outcomes have been considered as critical determinants of health and health service outcomes [8]. Although there is a growing body of literature documenting health disparities, the periurban-urban divide is still not fully understood. [9].

In the Ashanti region, Ghana, the burden of Hypertensive Disorders in Pregnancy (HDPs) is high affecting over a fifth (21.4%) of pregnant women. This may be attributed to poor quality of antenatal care (ANC) services or poor maternal knowledge on HDPs etc [10]. Similarly, the high burden of HDPs would translate into poor health outcomes such as high maternal and neonatal mortalities. A recent study also found HDPs to be common in the patterns of obstetric emergencies by 20.57%. [11]. Some studies have indicated high rural-urban disparities amongst hypertensive patients in Ghana [12–14]. Nonetheless there is a paucity of evidence of the health outcomes of HDPs and disparities in the ANC services offered to pregnant women, particularly those within the peri-urban (settlements along the peripheries of the urban communities) settings in the region.

Thus, this study sought to assess the differences in the quality of Ante-natal Care (ANC) and the outcomes of Hypertensive Disorders in Pregnancy (HDPs) from selected health facilities in the Ashanti region of Ghana, so as to help promote targeted interventions to improve HDPs outcomes in pregnancy. Research finding on these outcomes would provide insight on the management of hypertensive disorders in order to address the high burden of the disease in the region. It would provide adequate evidence for resource distribution and coordination in the face of limited health resources to improve health service gaps that exist amongst facilities in the region.

# Methods

## Study design and site

This study is part of a cohort study which assessed the hypertensive disorders amongst pregnant women and the birth outcomes. It employed a pretested structured questionnaire to obtain data from pregnant women who were seeking care at different levels of the health care delivery system in the Ashanti region (Ghana). These health facilities (hospitals) were located in four different districts (municipals) and included, Ejisu Government hospital (primary level, Ejisu Municipal), University hospital (primary level/quasi-government, Oforikrom Municipal), Kumasi South Hospital (secondary level, Asokwa Municipal) and Komfo Anokye Teaching hospital, KATH (tertiary level, Kumasi Metropolis,).

## Sample and sampling

This study included pregnant women aged 18years and above who had been admitted for delivery and those who had delivered at the hospital during the period of the study. Pregnant women who were not due for delivery and those with co-morbid conditions were excluded. Using the Yamane's formula at a significance level of 5% [15], an estimated sample size of 440

pregnant women was adequate. Making allowance for non-response rate, a minimum of 500 pregnant women were recruited. The number of pregnant women from each facility was determined in proportion to the documented average monthly delivery at each hospital. Hence, 105 (21%), 125(25%), 120(24%) and 150(30%) mothers were recruited from the Ejisu district hospital, university hospital, Kumasi South Hospital and KATH respectively. After explaining the purposes and benefits of the study to eligible women, signed informed consent obtained. Eligible women were randomly selected using an online number generator. All ethical issues were adhered to, and mothers were assured of privacy and confidentiality.

## Development and validation of questionnaire

The questionnaire was developed after reviewing several literatures to ensure that the items capture a meaningful construct to have causality to the outcome of interest. The questionnaire was piloted at the Manhyia Government hospital, which was not part of the study. Information was collected on their socio-demographic characteristics, proportions of HDPs, level of knowledge, quality of ANC and the birth outcomes.

Cronbach alpha test was used to assess the reliability of the construct of composite variables such as the level of knowledge. The score for level of knowledge was 0.73 indicating that the composite variable is reliable. The level of knowledge was assessed as an 18-item composite variable. For a correct response a code of 1 was assigned and a code of 0 for a wrong response. Therefore, the possible maximum and minimum scores for a participant were 18 and 0 respectively. Where a participant scored less than 5 then the level of knowledge of the participant was graded as poor, scoring between 5 and 9 was considered satisfactory, 10 to 14 as good and scores of 15 to 18 were considered to be excellent.

The quality of ANC was assessed using a formed table as part of the questionnaire which was adapted from the WHO quality indices for ANC 2018. It was measured as a composite variable made up of patient satisfaction, availability of staff, the routine vital checks and education during ANC visit. This helped to evaluate the quality of ANC that could influence the early detection and management of hypertensive disorders in pregnancy in line with existing protocols.

The dependent variable was the location of the health facilities where mothers sought care. Two hospitals were selected within Kumasi Metropolis (KATH and University Hospital) and the other two, from the peripheral settlements of the metropolis (Kumasi South Hospital and Ejisu Government Hospital), therein referred to as urban and peri urban respectively. These facilities were chosen because of their location in the region and the large coverage of patients they manage. A code of 1 and 0 was assigned to participants who sought care at the urban and periurban health facilities respectively. Thus, the dependent variable was measured as a binary outcome.

The independent variables included type of HDPs [chronic hypertension, chronic superimposed with preeclampsia, gestational hypertension and Pre-eclampsia], quality of ANC, knowledge on HDPs and the outcomes of pregnancy. HDPs, was measured as a mothers having blood pressure above 140/90mmHg beyond 20weeks gestation with or without history of hypertension or evidence of organ damage as confirmed and documented by the midwives or a doctor in the patient's records book.

## Data collection

A research team comprising medical officers, Pharmacists, Pharmacy house officers, research scientists and midwives was formed and oriented through series of meetings on the procedures and the objectives of the study. Each member had a specific role to play. The team was grouped

into four smaller groups with a leader for each hospital. All consenting women underwent a confidential face-to-face interview (where possible) in vernacular or English using the pre-tested questionnaire by a member of the research team. Their antenatal records were also reviewed for other information. The participants were allowed to withdraw from the study at any time if they so wished and reasons for withdrawal documented. All data was collected and accessed from 1st January to 30th April 2022.

## Data analysis

Data on the Excel were cross-checked, cleaned, coded, and were imported into Stata 14.1 (Stata Corp, Texas, USA) for statistical analysis. A univariate analysis employing the Pearson's Chi square test and univariable logistic regression analysis was carried out to assess the factors associated with location of the healthy facility. Variables which met the criteria for inclusion into the multivariable logistic regression model one at a time to assess independent association with the outcome of interest. The univariate and multivariable analysis were expressed as crude and adjusted Odds ratio with their respective 95% Confidence Intervals (CIs) and p-values. Statistical significance was considered at $p < 0.05$.

## Ethics approval and consent to participate

The KNUST Committee for Human Research and Ethics (Reference–CHRPE/AP/053/21) and the Komfo Anokye Teaching Hospital Institutional Review Board (KATH IRB/AP/018/21) provided the ethical clearance for the conduct of the study. In addition, approvals were given from the Research board of KNUST hospital (UH/51/Vol.1), Ejisu Government Hospital (EJ/EGH/EJ-209/20) and The Kumasi South Hospital (KSH. /RESH-50). All study procedures were conducted in line with related guidelines and regulations. Informed consent was sought from all pregnant women before enrolment in the study. Participants were assured of their right to be informed about the findings of the study.

## Results

### Demographic characteristics of participants

A total of 500 pregnant women were included in this study. The background characteristics of respondents recruited from peri urban facilities compared to those from urban facilities are shown in Table 1. The age group, ethnicity, education, religion, and occupation were significantly different for women from peri urban and urban facilities. Women from urban facilities were more likely to be older, Akans or Bonos (p = 0.04), more educated (p<0.01), married (p = 0.54), Christian (p<0.01), managing their private business (p<0.01) or take more time to travel to the health facility (p<0.01) compared to their counterparts recruited from the peri-urban facilities.

### Comparing the health outcomes between peri urban and urban health facilities

Health outcomes for women from urban and peri-urban health facilities differed significantly with regards to the hypertensive state, type of HDPs, quality of ANC, gestational age at delivery, and knowledge of HDPs among the women, Table 2. Women attending urban health facilities were more likely to be hypertensive ($\mu^2 = 126.4$; p<0.001), have chronic hypertension with superimposed pre-eclampsia (p<0.001), have good quality ANC ($\mu^2 = 41.28$; p< 0.001), deliver full term ($\mu^2 = 4.83$; p = 0.028), and have excellent knowledge of HDPs ($\mu^2 = 227.65$;

**Table 1. Comparison of demographic characteristics of respondents from urban and periurban health facilities.**

| Variable | Category | Women from peri-urban health facilities n (%) | Women from urban health facilities n (%) | P-value |
|---|---|---|---|---|
| Age group (years) | 18–24 | 39 (16.96) | 60 (22.22) | **0.04**\* |
| | 25–29 | 88 (38.26) | 86 (31.85) | |
| | 30–34 | 63 (27.39) | 60 (22.22) | |
| | 35–39 | 34 (14.78) | 44 (16.30) | |
| | 40–44 | 6 (2.61) | 20 (7.41) | |
| Ethnicity | Akans | 88 (38.43) | 206 (75.93) | **< 0.01**\* |
| | Bono | 6 (2.62) | 16 (5.93) | |
| | Ewe | 17 (7.42) | 13 (4.81) | |
| | Ga | 4 (1.75) | 4 (1.48) | |
| | Northerner | 40 (17.47) | 28 (10.37) | |
| | Other | 74 (32.31) | 4 (1.48) | |
| Educational status | No education | 54 (23.68) | 26 (9.63) | **< 0.01**\* |
| | Primary | 23 (10.09) | 7 (2.59) | |
| | JHS | 51 (22.37) | 56 (20.74) | |
| | SHS | 53 (23.25) | 76 (28.15) | |
| | Tertiary | 47 (20.61) | 105 (38.89) | |
| Marital status | Single | 50 (21.83) | 69 (25.65) | 0.54 |
| | Married | 178 (77.73) | 198 (73.61) | |
| | Divorced | 1 (0.44) | 2 (0.74) | |
| Religion | Christian | 169 (75.11) | 240 (89.22) | **< 0.01**\* |
| | Moslem | 26 (11.56) | 28 (10.41) | |
| | Others | 30 (13.33) | 1 (0.37) | |
| Occupation | Govt Employee | 37 (16.44) | 79 (29.37) | **< 0.01**\* |
| | Housewife | 33 (14.67) | 27 (10.04) | |
| | Private Business | 155 (68.89) | 163 (60.59) | |
| Proximity to health facility | < an hour | 130 (56.52) | 46 (17.10) | **< 0.01**\* |
| | > an hour | 100 (43.48) | 223 (82.90) | |
| Heard of HDPs | No | 5 (2.20) | 4 (1.52) | 0.57 |
| | Yes | 222 (97.80) | 260 (98.48) | |

\* = Statistically significant.

p< 0.001) compared to women receiving care in peri-urban health facilities. The pregnancy and delivery outcomes did not differ significantly between the two groups.

## Comparing the utilisation and quality of ANC by women visiting urban and periurban health facilities

The quality and utilisation of ANC by women from urban and peri-urban health facilities differed significantly with regards to the number of and their first ANC visits, attitude of healthcare provider during ANC (HP) and their partner support, Table 3. Women attending urban health facilities were more likely to have their first antenatal visit in the first trimester ($\mu^2 = 44.12$; p<0.001), lesser ANC visits ($\mu^2 = 290.13$; p< 0.001), to have received attitudes from their HP during ANC ($\mu^2 = 20.12$; p< 0.001), and also enjoy excellent partner support ($\mu^2 = 38.44$; p< 0.001) compared to women receiving care in peri-urban health facilities. Their plans 'to get pregnant' did not differ significantly between the two groups.

**Table 2. Health outcome differences between urban and peri urban health facilities.**

| Variable | Category | Location of Hospitals | | $\mu^2$ | p-value |
|---|---|---|---|---|---|
| | | Periurban | Urban | | |
| Hypertensive status | Normotensive | 205(89.1%) | 109(40.4%) | 126.4 | **< 0.001** |
| | Hypertensive | 25 (10.9%) | 161(59.6%) | | |
| Type of HDPs | Chronic hypertension | 9 (36.0%) | 42(26.1%) | | **< 0.001** |
| | Chronic hypertension with superimposed pre-eclampsia | 3 (12.0%) | 85(52.8%) | | |
| | Pre-Eclampsia | 8 (32.0%) | 26 (16.2%) | | |
| | Gestational hypertension | 5 (20.0%) | 8 (5.0%) | | |
| Quality of ANC | Poor | 60 (26.4%) | 15 (5.6%) | 41.28 | **< 0.001** |
| | Good | 167(73.6%) | 252 (94.4%) | | |
| Outcome of pregnancy | Live Birth | 219(95.6%) | 256(94.8%) | 0.18 | 0.67 |
| | Stillbirth | 10 (4.4%) | 14 (5.2%) | | |
| Delivery outcome | SVD | 146 (63.8%) | 188 (70.2%) | 2.29 | 0.131 |
| | C/S | 83 (36.2%) | 80 (29.9%) | | |
| Term | Full term | 211 (91.7%) | 228 (85.4%) | 4.83 | **0.028** |
| | Preterm | 19 (8.3%) | 39 (14.6) | | |
| Knowledge of mothers | <6 (Poor) | 58 (25.2%) | 11 (4.1%) | 227.65 | **< 0.001** |
| | 6–9 (Satisfactory) | 110(47.8%) | 21 (7.9%) | | |
| | 10–14 (Good) | 39 (17.0%) | 39 (14.6%) | | |
| | 15–18 (Excellent) | 23 (10.0%) | 196(73.4%) | | |

* = Statistically significant.

## Comparing the assessments of health indices between mothers recruited from urban and periurban facilities

The hypertensive status of mothers, utilization of ANC, their knowledge on HDPs and the proximity of mothers to the health facilities significantly differed between the two settings. It was realized that mothers from the urban areas were more likely to be hypertensive (AOR = 0.10, CI (95%) = 0.02–0.68), less likely to have their first ANC visit during their third trimester (AOR = 0.03, CI (95%) = 0.00–0.35), likely to have knowledge on HDPs (AOR = 6.84, CI (95%) = 1.53–30.47) and travelled over one hour to the health facilities (AOR = 3.78, CI (95%) = 1.48–9.64).

## Discussion

This study sought to assess differences in quality of ANC and outcomes of HDPs among pregnant women seeking obstetric care in urban and peri urban health facilities in the Ashanti region of Ghana. Women receiving obstetric care in urban health facilities differed significantly from those attending peri-urban health facilities in terms of sociodemographics, health outcomes on hypertensive disease states, quality of ANC, timing of delivery and knowledge on HDPs.

From the findings of this study, it was realized that the burden of hypertensive disorders in pregnancy varied considerably between periurban health facilities and urban health facilities. The burden of HDPs was 10.9% and 59.6% amongst mothers seeking antenatal care at the periurban and urban health facilities respectively. (Table 2). This means that nearly one tenth of the mothers who sought care at the periurban centers were experiencing HDPs. This results is similar to a study which had indicated that, the growing prevalence of hypertension in low- and middle-income nations, especially those in sub-Saharan Africa, has been attributed in part

**Table 3. ANC Utilization between mothers visiting periurban and urban hospitals.**

| Variable | Category | Periurban | Urban | $\mu^2$ | p-value |
|---|---|---|---|---|---|
| First Antenatal visit | First Trimester | 123 (53.7) | 219 (81.4) | 44.12 | **< 0.01**[*] |
| | Second Trimester | 93 (40.6) | 44 (16.4) | | |
| | Third Trimester | 13 (5.7) | 6 (2.2) | | |
| | | | | | |
| Number of ANC visit | 1 – 3times | 25 (11.0) | 236 (87.4) | 290.13 | **< 0.01**[*] |
| | 4 – 6times | 98 (43.2) | 22 (8.2) | | |
| | Above 7times | 104 (45.8) | 12 (4.4) | | |
| Attitude of HP during ANC | Excellent | 54 (23.9) | 93 (34.8) | 20.12 | **< 0.01**[*] |
| | Good | 114 (50.4) | 139 (52.1) | | |
| | Satisfactory | 14 (6.2) | 2 (0.8) | | |
| | Poor | 44(19.5) | 33(12.4) | | |
| Influence to attend ANC | Family | 16 (7.0) | 35 (13.1) | 15.36 | **< 0.01**[*] |
| | Husband | 30 (13.0) | 12 (4.5) | | |
| | Self | 184 (80.0) | 221 (84.4) | | |
| Partner support | Excellent | 42 (18.3) | 108 (40.2) | 38.44 | **< 0.01**[*] |
| | Good | 65 (28.3) | 59 (21.9) | | |
| | Satisfactory | 94 (40.9) | 94 (34.9) | | |
| | Poor | 29 (12.6) | 8 (3.0) | | |
| Plan to have pregnancy | No | 22 (9.9) | 23 (8.6) | 0.23 | 0.63 |
| | Yes | 201 (90.1) | 244 (91.4) | | |
| NHIS status | Unregistered | 14 (6.2) | 1 (0.4) | 13.98 | **< 0.01**[*] |
| | Registered | 231 (93.8) | 266 (99.6) | | |

* = Statistically significant.

to urbanization (SSA) [16]. Although there is no single, applicable definition for urbanization, the phenomena frequently reflects shifts in political, social, and economic dynamics, which have a significant impact on lifestyle choices such sources of income, diet, transportation, and family structure [17]. Rapid urbanization is thought to contribute to the development of hypertension by encouraging poorer eating habits, sedentary lifestyles, alcohol and tobacco use, the development of obesity, and exposure to more psychological stressors. This is especially true if it is unplanned or poorly planned, as is frequently the case in much of West Africa [18]. However, in Ghana, rural or some parts of periurban areas are generally characterised by a more active lifestyle with farming as the major occupation, a diet that contains less processed foods, and less exposure to environmental pollution compared to urban areas. Urban socioeconomic and lifestyle changes that may increase the incidence of its risk factors like overweight, obesity, smoking, and diabetes mellitus are possible explanations for a higher prevalence of hypertension in those places [19]. Dietary variables, particularly the consumption of processed high-salt meals, may potentially be a factor [20]. Other reasons may include redefining cultural identity, psychosocial stress in urban dwellers brought on by financial stress, and moving away from conventional coping mechanisms, such as social support from extended family [21]. The exhaust fumes from motorized cars and businesses, which tend to worsen air pollution in urban areas, may also have a role [22]. However, the greater accessibility to medical facilities in urban as opposed to periurban areas may also be a factor in the higher reported prevalence in these places [19].

Knowledge of an individual on a particular health conditions influences the health seeking behavior of the individual [23]. Thus, awareness on HDPs has been assessed as efforts at

preventing HDPs burden. It was realized that about 73.4% and 10.0% of the mothers in the urban and periurban had excellent knowledge of HDPs respectively. This study showed that the level of knowledge was significantly high amongst pregnant women in urban health facilities compared to periurban. It confirms the Bello et al., study which found that mothers from the urban areas were more likely to be knowledgeable on HDPs compared to mothers from the periurban area. It was also realized that urban mothers were 6.8 times more likely to be knowledgeable on HDPs compared to mothers from the periurban areas (Table 4). Tran et al., (2011)revealed in a study that level of education of mothers at the urban areas were higher than mothers from the peri-urban, hence it is not surprising that the level of knowledge is higher amongst mothers visiting urban health facilities [24]. This difference in level of knowledge could influence health seeking behavior of women in the periurban setting which may also affect their pregnancy outcomes. Educational campaigns on HDPs during ANC should be given to heighten in health facilities within the periurban areas.

The quality of ANC was assessed as a composite variable assessing the patient's perception on the antenatal care services they receive at the various facilities. The WHO in 2016 recommended activities for the ANC model and patients perception on these activities were assessed [25]. It was observed that majority of mothers perceived the quality of Antenatal care to be good in both periurban and urban health facilities in the Ashanti region (Ghana). The proportion of mothers who considered the quality of ANC to be good in the urban was 94.8% which was higher than the 73% of mothers who considered the quality of ANC to be good in the periurban health facilities. It was also realized that mothers in the urban communities 3.8 times more likely to travel over one hour to health facilities for their ANC visits compared to their counterparts in the periurban areas but periurban mothers may utilize ANC services more. (Table 4) There is therefore disparity in the utilisation and quality of care received at the various health facilities. The antenatal care services offer the opportunity for early detection of pregnancy related complications through frequent and regular screening. Additionally, the antenatal care services also helps in promoting disease prevention through education and counselling [26]. As such this difference in the perceived quality of antenatal care services between the urban and periurban health care facilities in the Ashanti region could lead to disparity in obstetric outcomes. A contributing factor to the witnessed difference may be due to the high turnover of health professionals in the periurban areas compared to the urban areas in Ghana [27]. Thus, health professionals in the periurban health facilities are sometimes overwhelmed and that leads to burnout and subsequent poor performance which impacts on quality of care. Health professionals should be encouraged to take up their roles in periurban health facilities when their transfers are due.

It was observed that the proportion of stillbirth did not vary between the periurban and urban hospitals. It was expected that the proportion of stillbirth would be high in the urban hospitals because they are often overburdened and overwhelmed by complicated cases which may sometimes lead to a stretch on the health professionals.

From the findings of the study, it was also realized that the burden of caesarean section was 36.2% in the periurban 29.9% at the urban health facilities. However, this difference was found to be statistically insignificant. We conclude that the mode of delivery of pregnancy for mothers was similar for both periurban and urban hospital. It was realized that the burden of caesarean section was very high amongst mothers in both locations in the Ashanti region compared to the national burden of caesarean section deliveries. Findings from the 2014 Ghana Demographic and Health Survey reveals the national burden to be 18.5% [28]. Thus, the burden of caesarean section in periurban health facilities in the Ashanti region was nearly twofold that of the national burden and the burden of caesarean section had increased by about 61% in the urban health facilities in the Ashanti region compared to the national burden. This high

**Table 4. Assessment of health indices between urban and periurban facilities.**

| Variable | Category | COR | CI (95%) | AOR | CI (95%) |
|---|---|---|---|---|---|
| Age group (years) | 18–24 | Ref | | Ref | |
| | 25–29 | 0.64 | 0.38–1.05 | 0.81 | 0.24–2.74 |
| | 30–34 | 0.62 | 0.36–1.06 | 1.27 | 0.32–5.02 |
| | 35–39 | 0.84 | 0.46–1.54 | 1.68 | 0.49–7.17 |
| | 40–44 | 2.17 | 0.80–5.87 | 0.15 | 0.01–2.09 |
| Ethnicity | Asante | Ref | | Ref | |
| | Bono | 1.14 | 0.43–3.02 | 1.05 | 0.13–8.53 |
| | Ewe | 0.33 | 0.15–0.70 | **6.20** | **1.11–34.61*** |
| | Ga | 0.43 | 0.10–1.76 | 0.82 | 0.06–11.57 |
| | Northerner | 0.30 | 0.17–0.52 | 1.14 | 0.24–5.44 |
| | Other | 0.02 | 0.01–0.07 | 0.09 | 0.01–0.73 |
| Educational status | No education | Ref | | Ref | |
| | Primary | 0.63 | 0.24–1.66 | 1.68 | 0.17–16.73 |
| | JHS | 2.28 | 1.25–4.17 | **11.71** | **2.63–52.14*** |
| | SHS | 2.98 | 1.66–5.34 | **7.41** | **1.83–30.03*** |
| | Tertiary | 4.64 | 2.60–8.29 | 1.19 | 0.21–6.78 |
| Religion | Christian | Ref | | Ref | |
| | Moslem | 0.76 | 0.43–1.34 | 1.13 | 0.20–6.41 |
| | Others | 0.02 | 0.00–0.17 | 0.00 | 0.00–27.14 |
| Occupation | Gov Employee | Ref | | Ref | |
| | Housewife | 0.38 | 0.20–0.73 | 0.20 | 0.02–1.96 |
| | Private Business | 0.49 | 0.31–0.77 | 0.60 | 0.12–2.92 |
| Proximity to health facility | < an hour | Ref | | Ref | |
| | > an hour | 6.30 | 4.18–9.50 | **3.78** | **1.48–9.64*** |
| Type of HDPs | Chronic hypertension | Ref | | Ref | |
| | Chronic hypertension with superimposed pre-eclampsia | 6.07 | 1.56–23.61 | 4.88 | 0.28–86.29 |
| | Pre-Eclampsia | 070 | 0.24–2.03 | 0.40 | 0.02–7.28 |
| | Gestational hypertension | 0.34 | 0.91–1.30 | 0.28 | 0.02–4.96 |
| | Normotensive | | | **0.10** | **0.02–0.68*** |
| First Antenatal visit | First Trimester | Ref | | Ref | |
| | Second Trimester | 0.27 | 0.17–0.40 | 1.15 | 0.38–3.48 |
| | Third Trimester | 0.26 | 0.01–0.70 | **0.03** | **0.00–0.35*** |
| | | | | | |
| Number of ANC visit | 1 – 3times | Ref | | Ref | |
| | 4 – 6times | 0.02 | 0.01–0.04 | **0.02** | **0.00–0.07*** |
| | Above 7times | 0.01 | 0.01–0.03 | **0.01** | **0.00–0.06*** |
| | | | | | |
| Attitude of HP during ANC | Excellent | Ref | | Ref | |
| | Good | 0.71 | 0.47–1.07 | 1.49 | 0.51–4.34 |
| | Satisfactory | 0.44 | 0.25–0.76 | 4.44 | 0.87–22.69 |
| | Poor | 0.08 | 0.02–0.38 | 0.14 | 0.00–14.56 |
| Influence to attend ANC | Self | Ref | | Ref | |
| | Husband | 0.33 | 0.17–0.67 | 1.02 | 0.18–5.88 |
| | Family | 1.82 | 0.98–3.40 | **18.22** | **3.45–96.07*** |
| Partner support | Excellent | Ref | | Ref | |
| | Good | 3.63 | 1.58–8.34 | 0.52 | 0.08–3.40 |
| | Satisfactory | 3.29 | 1.39–7.76 | 1.20 | 0.17–8.39 |

(*Continued*)

**Table 4.** (Continued)

| Variable | Category | COR | CI (95%) | AOR | CI (95%) |
|---|---|---|---|---|---|
|  | Poor | 9.32 | 3.94–22.03 | 0.60 | 0.08–4.53 |
| NHIS status | Unregistered | Ref |  | Ref |  |
|  | Registered | 17.48 | 2.28–134.0 | 2.95 | 0.00–1944.63 |
| Quality of ANC | Poor | Ref |  | Ref |  |
|  | Good | 6.03 | 3.32–10.98 | 3.96 | 0.97–16.13 |
| Term | Full term | Ref |  | Ref |  |
|  | Preterm | 1.90 | 1.06–3.39 | 1.54 | 0.35–6.78 |
| Knowledge of mothers | Poor | Ref |  | Ref |  |
|  | Satisfactory | 1.01 | 0.45–2.23 | 1.11 | 0.31–4.06 |
|  | Good | 5.27 | 2.41–11.53 | 1.85 | 0.34–10.19 |
|  | Excellent | 44.93 | 20.68–97.62 | **6.84** | **1.53–30.47**[*] |

Adjusted for all other variables shown.

[*] = Statistically significant COR = Crude Odds Ratio.

CI = Confidence interval AOR = Adjusted Odds Ratio.

burden of caesarean section was however, consistent with the findings of Prah et al., (2017) where it was realized the burden of caesarean section in a health facility was 26.9%. The attributable reasons for this high burden were the lack of diagnostic technologies in both urban, periurban and even tertiary health facilities that leads to delayed diagnosis and the increased request from mothers for caesarean section over spontaneous vaginal delivery [29]. There is the need to further investigate the factors that influence the high burden of caesarean section in the region.

Again, the proportion of babies who were delivered preterm was found to be higher in urban hospitals compared to the proportion of babies who were delivered preterm in periurban health facilities. The term preterm birth refers to neonatal birth before the 37 full weeks of gestation or measured as the delivery prior to 259days from the onset of the mother's last menses [30]. Preterm delivery was 8.3% in the periurban health facilities while preterm birth was high in the urban facilities at a rate of 14.6%. The burden of preterm in urban health facilities is similar to the national burden of 14.5% [30]. These observations may be due to the fact that the burden of hypertensive disorders in the urban facilities is more than five times that of the periurban facilities. Hypertensive disorders in pregnancy are also established to be associated with high preterm delivery due to possible foetal distress. This therefore accounts for the excess burden of preterm delivery in urban health facilities compared to the periurban. The consequences of preterm are the increased risk of low birth weight, perinatal asphyxia, perinatal deaths and high cost of parental care [31,32].

## Conclusion

There were differences in quality of ANC and health outcomes between periurban-urban health facilities in Ashanti region of Ghana. Mothers attending urban health facilities were likely to be hypertensive, receive quality ANC services, and have excellent knowledge on HDPs compared to women receiving care in peri-urban health facilities. The method of delivery and the outcome of birth did not statistically vary between the two groups. However, there was an increase in the proportion of preterm deliveries in urban compared to periurban. There is a need to target maternal care interventions to the periurban health facilities in order to improve obstetric health outcomes.

## Supporting information

**S1 Checklist. STROBE statement—checklist of items that should be included in reports of observational studies.**
(DOCX)

## Acknowledgments

We would like to acknowledge the hospital management, members of the research team, and all participants of the study.

## Author Contributions

**Conceptualization:** Pauline Boachie-Ansah, Afia Frimpomaa Asare Marfo, Edward Tieru Dassah.

**Data curation:** Pauline Boachie-Ansah, Joseph Attakora.

**Formal analysis:** Pauline Boachie-Ansah, Afia Frimpomaa Asare Marfo.

**Funding acquisition:** Pauline Boachie-Ansah.

**Investigation:** Pauline Boachie-Ansah, Ivan Eduku Mozu.

**Methodology:** Pauline Boachie-Ansah, Afia Frimpomaa Asare Marfo, Edward Tieru Dassah, Ivan Eduku Mozu, Joseph Attakora.

**Supervision:** Berko Panyin Anto.

**Validation:** Berko Panyin Anto.

**Writing – original draft:** Pauline Boachie-Ansah.

**Writing – review & editing:** Pauline Boachie-Ansah, Berko Panyin Anto, Afia Frimpomaa Asare Marfo, Edward Tieru Dassah, Ivan Eduku Mozu, Joseph Attakora.

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
