## [Decision Letter · Decision Letter 0]

5 Jul 2023

PONE-D-23-07721Quality of Antenatal Care (ANC) and Outcomes of Hypertensive Disorders in Pregnancy (HDPs) among Antenatal Attendees: A Comparison of Urban and Periurban Health Facilities in Ghana.PLOS ONE

Dear Dr. Boachie-Ansah,

Thank you for submitting your manuscript to PLOS ONE. After careful consideration, we feel that it has merit but does not fully meet PLOS ONE’s publication criteria as it currently stands based on my own assessment and the reviewers' report. Therefore, we invite you to submit a revised version of the manuscript that addresses the points raised during the review process. I recommend you address all the comments raised by the reviewers found below this letter.

We look forward to receiving your revised manuscript.

Kind regards,

Desmond Kuupiel, PhD

Academic Editor

PLOS ONE

Journal Requirements:

2. "In your Data Availability statement, you have not specified where the minimal data set underlying the results described in your manuscript can be found. PLOS defines a study's minimal data set as the underlying data used to reach the conclusions drawn in the manuscript and any additional data required to replicate the reported study findings in their entirety. All PLOS journals require that the minimal data set be made fully available. For more information about our data policy, please see http://journals.plos.org/plosone/s/data-availability.

"PBA received a local scholarship from the Ghana National Petroleum Corporation (GNPC) Foundation, Ghana (GNPC/FDN/C4/KNUST/18)

Sponsors played no role in the study design or manuscript

www.gnpcghana.com/scholarship.html"              

Reviewers' comments:

Reviewer's Responses to Questions

**Comments to the Author**

1. Is the manuscript technically sound, and do the data support the conclusions?

Reviewer #1: Partly

Reviewer #2: Yes

2. Has the statistical analysis been performed appropriately and rigorously? 

Reviewer #1: No

Reviewer #2: Yes

3. Have the authors made all data underlying the findings in their manuscript fully available?

Reviewer #1: Yes

Reviewer #2: Yes

4. Is the manuscript presented in an intelligible fashion and written in standard English?

Reviewer #1: No

Reviewer #2: Yes

5. Review Comments to the Author

Reviewer #1: QUALITY OF ANTENATAL CARE AND OUTCOMES OF HYPERTENSIVE

DISORDERS IN PREGNANCY AMONG ANTENATAL ATTENDEES: A COMPARISON OF URBAN AND PERIURBAN HEALTH FACILITIES IN GHANA

Below are my comments for the above article.

Abstract

The results, as stated in the abstract, needed to provide more data. It reads like a discussion.

I suggest the authors report the data to support the statements.

HPDs abbreviation in the abstract was not correctly placed.

Please proofread the manuscript.

Introduction

Lines 50 to 51: Provide references.

Lines 60 to 61: The references are more than 10 years and cannot be the current evidence.

Please be consistent with HDP usage throughout the manuscript. Sometimes Hypertensive disorders or hypertensive disorders of pregnancy is used.

When a region is used, indicate the country.

The aim stated in the abstract, and the introduction are not consistent. This made it difficult to review the manuscript.

Method:

18 years and above was the only inclusion criteria? Did the participants receive any incentives?

Both independent and dependent variables were not clearly defined. Which diagnoses were included in HDP? Chronic hypertension, gestational hypertension, preeclampsia?

Hypertension before 20 weeks gestation is considered pre-existing/chronic hypertension. It is not clear the criteria that support “blood pressure above 140/90 beyond 16 weeks gestation” as HDP. Clearly defining the variables could help.

The data analysis section needed to be revised.

Results and Discussion

Due to missing information on the method, and inconsistent objectives, I could not evaluate the results and discussion adequately.

Reviewer #2: Manuscript ID: PONE-D-23-07721

Title: Quality of Antenatal Care (ANC) and Outcomes of Hypertensive Disorders in

Pregnancy (HDPs) among Antenatal Attendees: A Comparison of Urban and Periurban Health Facilities in Ghana

Date: 7 May 2023

General comment

This topic of the study, which aimed to compare the difference ANC quality and outcome of HDP among ANC attendees and the sample size is also fine, sounds good. The manuscript was also technically sound, and the data supported the conclusions.

Comments for the specific area of improvement

In the abstract: Method: Line 25: it says...... involving 500 (54%, n=270) urban and

26 (46%, n=230) periurban pregnant women was conducted. Why the sample size differ among the areas (Periurban and urban)

In the introduction line 53: it says various studies and cited by only one citation? Authors should address this

Line 54: This is relevant to policy as rural and peri urban areas including health facilities have worse health outcomes compared to urban areas (Zeng et al., 2015). If so, what is the important of your study currently, meant you knew that as health outcome in rural and periurban is worse than urban. Revise it

The same to line 63, this statement make your study insignificant, unless remove it.

Line 67 -68: Perhaps their health outcomes may be better than the rural settlements due to their proximity to the urban areas. This is seems copy paste from a literature and it is implication of other study results. Unnecessary

Method

Study design and site

Line 83-90: you were compared study participants from periurban and urban one: but how could you control the overwhelming of the advanced hospitals over the health facilities found at periurban; meant if women referred to the advanced hospital from periurban or choosing advanced hospital for advanced care by their selves? Any mechanism to oversee such problem??

Sample and sampling

Line 93: Why you had selected women aged 18 and above? Any reason behind? Why not 15 years old?

Line 95: how it come 440 and then 500?? Show with steps?? How much is the non-response rate?

Line 115-117: what is your reference to measure knowledge with this cut points?? Is it appropriate to measure knowledge with such cut points?? Knowledge is continuous variable. It needs explanations

Line 174: table2: it says quality ANC ===poor and good; what is your reference for?

Discussion

The discussion is shallow in general

Line 190-193=== this statement controversial to your topic, it makes incomparable, as stated in the study design and site

Line 251-253: scientifically not sound, meant you compared with figures not statically value

Acknowledgement

Line 292: is the supervisor part of the authorship or not? If he is part of the authorship, no need of special acknowledgement

Generally I accepted this manuscript with minor revision.

Thanks

6. PLOS authors have the option to publish the peer review history of their article (what does this mean?). If published, this will include your full peer review and any attached files.

Reviewer #1: No

Reviewer #2: **Yes: **Mebrahtu Kalayu Chekole(MPH/RH)

<quillbot-extension-portal></quillbot-extension-portal>

---

## [Author Response · Author response to Decision Letter 0]

24 Aug 2023

Reviewer 1

Comment : The results, as stated in the abstract, needed to provide more data. It reads like a discussion.

I suggest the authors report the data to support the statements.

Response: The results section of the abstract has been modified accordingly.

Comment: HPDs abbreviation in the abstract was not correctly placed.

Response: This has been rectified

Comment: Please proofread the manuscript.

Response: The manuscript has been proofread for errors

Comment: Lines 60 to 61: The references are more than 10 years and cannot be the current evidence.

Response: This reference has been replaced.

Comment: Please be consistent with HDP usage throughout the manuscript. Sometimes Hypertensive disorders or hypertensive disorders of pregnancy is used.

Response: The manuscript has been reviewed to ensure that all HDPs abbreviations are consistent.

Comment: When a region is used, indicate the country.

Response: This has now been indicated throughout the manuscript.

Comment: The aim stated in the abstract, and the introduction are not consistent. This made it difficult to review the manuscript.

Response: Although both paragraphs meant the same, they have been synchronized to provide better clarity.

Comment: 18 years and above was the only inclusion criteria? Did the participants receive any incentives?

Response: This study included pregnant women above the ages of 18years who had been admitted for delivery and those who had delivered at the hospital during the period of the study.

Participants were not given any incentives.

Comment: Both independent and dependent variables were not clearly defined. Which diagnoses were included in HDP? Chronic hypertension, gestational hypertension, preeclampsia?

Response: The dependent variable was the location of the health facility, that is either urban or periurban health facility.

The independent variables were the quality of ANC, outcome of pregnancy, type of HDPs and knowledge of mothers on HDPs

For this study, HDPs included Chronic hypertension, chronic superimposed with preeclampsia, gestational hypertension and 

Pre-eclampsia.

Diagnosis was done using the Ghana Standard Treatment Guidelines (STG 2017) and the American College Of 

Obstetricians and Gynaecologists guidelines 222.

Comment: Hypertension before 20 weeks gestation is considered pre-existing/chronic hypertension. It is not clear the criteria that support “blood pressure above 140/90 beyond 16 weeks gestation” as HDP. Clearly defining the variables could help.

Response: The section has been revised accordingly.

Comment: Hypertension before 20 weeks gestation is considered pre-existing/chronic hypertension. It is not clear the criteria that support “blood pressure above 140/90 beyond 16 weeks gestation” as HDP. Clearly defining the variables could help.

Response: The section has been revised accordingly.

Reviewer 2 

Comment: In the abstract: Method: Line 25: it says...... involving 500 (54%, n=270) urban and

26 (46%, n=230) periurban pregnant women was conducted. Why the sample size differ among the areas (Periurban and urban)

Response: This manuscript forms a part of cohort study that was conducted. As such the categories urban and periurban were not the outcome of interest for the conduct of the study. Therefore, these categories did not influence the sample size determination nor the sampling approach. 

However, analysis of the study revealed interesting findings of health outcome disparities between the urban and periurban participants which the authors of this manuscript believe would contribute to the knowledge of high HDPs in Ghana.

Comment: In the introduction line 53: it says various studies and cited by only one citation? Authors should address this.

Response: This has been rectified

Comment: Line 54: This is relevant to policy as rural and peri urban areas including health facilities have worse health outcomes compared to urban areas (Zeng et al., 2015). If so, what is the important of your study currently, meant you knew that as health outcome in rural and periurban is worse than urban. Revise it

Response: This has been rectified

Comment: The same to line 63, this statement make your study insignificant, unless remove it.

Response: This sentence has been removed.

Comment: Line 67 -68: Perhaps their health outcomes may be better than the rural settlements due to their proximity to the urban areas. This is seems copy paste from a literature and it is implication of other study results. Unnecessary

Response: This sentence has removed.

Comment: Line 83-90: you were compared study participants from periurban and urban one: but how could you control the overwhelming of the advanced hospitals over the health facilities found at periurban; meant if women referred to the advanced hospital from periurban or choosing advanced hospital for advanced care by their selves? Any mechanism to oversee such problem??

Response: For the cohort study, participants consented to part of the study agreed to visit and deliver at the specified hospital during the duration of study. 

As per the protocol, participants who were referred for another hospital or relocated from the catchment area during the conduct of the study were considered to have deviated or lost to follow up. Such participants are not included in the analysis. Therefore, participants who were part of this study continued with their visits and delivered at the specific health facility during the study period.

Comment: Line 93: Why you had selected women aged 18 and above? Any reason behind? Why not 15 years old?

Response: In Ghana, an 18-year-old is legally considered a full-fledged adult with rights and responsibilities. 

Therefore, such individuals (18years and above) can decide for themselves whether to agree to participate in the study or not.

For those younger than 18years, consent to participate in the study must be sought from their parents or guidance, which might cause a bit of delay.

Comment: Line 95: how it come 440 and then 500?? Show with steps?? How much is the non-response rate?

Response: The addition of 60participants was to take care of possible non-response from participants from the four facilities which could affect the external validity of the sample.

Comment: Line 115-117: what is your reference to measure knowledge with this cut points?? Is it appropriate to measure knowledge with such cut points?? Knowledge is continuous variable. It needs explanations.

Response: There were no references to the measure of knowledge as the items for assessing knowledge were authors own construct after several literature review and pretesting of the measurement tool.

Comment: Line 174: table2: it says quality ANC ===poor and good; what is your reference for?

Response: There were no references to the measure of ANC. However, the measurements were done in line with the WHO requirements for ANC quality indices. This was used to obtain a score and participants who scored more than half of the total were considered to have had good ANC.

Comment: The discussion is shallow in general

Response: This has been improved.

Comment: Line 190-193=== this statement controversial to your topic, it makes incomparable, as stated in the study design and site

Response: This statement has been revised.

Comment: Line 251-253: scientifically not sound, meant you compared with figures not statically value

Response: This has been revised

Comment: Line 292: is the supervisor part of the authorship or not? If he is part of the authorship, no need of special acknowledgement.

Response: This has been rectified.

---

## [Decision Letter · Decision Letter 1]

31 Oct 2023

Quality of antenatal care and outcomes of Hypertensive Disorders in Pregnancy among antenatal attendees: A comparison of urban and periurban health facilities in Ghana

PONE-D-23-07721R1

Dear Dr. Boachie-Ansah,

We’re pleased to inform you that your manuscript has been judged scientifically suitable for publication and will be formally accepted for publication once it meets all outstanding technical requirements.

Kind regards,

Khin Thet Wai, MBBS, MPH, MA

Academic Editor

PLOS ONE

Additional Editor Comments (optional):

Reviewers' comments:

Reviewer's Responses to Questions

**Comments to the Author**

1. If the authors have adequately addressed your comments raised in a previous round of review and you feel that this manuscript is now acceptable for publication, you may indicate that here to bypass the “Comments to the Author” section, enter your conflict of interest statement in the “Confidential to Editor” section, and submit your "Accept" recommendation.

Reviewer #2: All comments have been addressed

2. Is the manuscript technically sound, and do the data support the conclusions?

Reviewer #2: Yes

3. Has the statistical analysis been performed appropriately and rigorously? 

Reviewer #2: Yes

4. Have the authors made all data underlying the findings in their manuscript fully available?

Reviewer #2: Yes

5. Is the manuscript presented in an intelligible fashion and written in standard English?

Reviewer #2: Yes

6. Review Comments to the Author

Reviewer #2: Authors have addressed all issues raised during the reviewing process of the manuscript. publication of this manuscript will contributed to evidence the scientific community.

7. PLOS authors have the option to publish the peer review history of their article (what does this mean?). If published, this will include your full peer review and any attached files.

Reviewer #2: No

---

## [Editor Report · Acceptance letter]

23 Nov 2023

PONE-D-23-07721R1 

Quality of antenatal care and outcomes of Hypertensive Disorders in Pregnancy among antenatal attendees: A comparison of urban and periurban health facilities in Ghana 

Dear Dr. Boachie-Ansah:

I'm pleased to inform you that your manuscript has been deemed suitable for publication in PLOS ONE. Congratulations! Your manuscript is now with our production department. 

Kind regards, 

on behalf of

Dr. Khin Thet Wai 

Academic Editor

PLOS ONE